# Projected Effects of a Deep Excavation Pit on the Existing Metro Tunnel and Findings of Geotechnical Monitoring: A Comparative Analysis

**Armen Z. Ter-Martirosyan** [1] [iD], **Valery P. Kivliuk** [2], **Ilya O. Isaev** [1,2] **and Victoria V. Rud** [1,*]

1   Department of Soil Mechanics and Geotechnical Engineering, National Research Moscow State Civil Engineering University, 26, Yaroslavskoye Shosse, 129337 Moscow, Russia; gic-mgsu@mail.ru (A.Z.T.-M.); isaevio@mosinzhproekt.ru (I.O.I.)

2   Department of Impact Assessment and Emergency Response Measures in the "Mosinzhproekt" Joint Stock Company, 10, Hodynsky Blvd, 125252 Moscow, Russia; kivliuk.vp@mosinzhproekt.ru

*   Correspondence: victoriadll@yandex.ru

**Abstract:** With the evolution of modern cosmopolitan cities, subterranean spaces have developed in dense urban environments. Hence, new metro tunnels often intersect with those in operation. The top-priority task of designers is to evaluate the effect of new construction projects. The experience accumulated in this field should contribute to the design of a realistic geotechnical model to simulate long-term displacements in the future. This paper includes a backward analysis of a design scheme developed for a tunnel construction area above an existing tunnel with a 10.3 m diameter, according to the results of geotechnical monitoring performed in PLAXIS 2D. The authors identified the optimum combination of the distance from the tunnel bottom to the lower boundary of the design model, the soil model, and tunnel lining stiffness. The authors derived regression equations describing vertical and horizontal displacements of the tunnel at the stage of excavation to the elevation datum as the excavation pit bottom. These equations can be applied to preliminarily predict the displacements of the tunnel depending on geometrical parameters at the initial design stage. Geometrical parameters include the distance from the tunnel to the excavation pit, the depth of the tunnel from the surface to the crown, the depth of the designed excavation pit, and the distance from the bottom of the excavation pit to the bottom of the tunnel. In addition, the effect of the Muir–Wood coefficient on the vertical displacements of the tunnel was investigated. This work found a reduction in the stiffness of the bearing structure of the tunnel and an increase of 4.8% in deformations on average when this coefficient was considered.

**Keywords:** automated geotechnical monitoring; numerical analysis; reduced second moment of area; multiple linear regression

## 1. Introduction

In recent years, newly designed tunnels have often intersected with existing ones due to the intensive development of the Moscow metro. A pre-existing structure is subjected to the effect of a new construction project, e.g., the engineering construction cases below (deep drilling tasks) and above metro tunnels (construction works near the ground surface). In the second case, additional tunnel deformations develop upwards, towards the excavation work, as a result of the removal of a large mass of soil and the decompaction of the soil layer on the excavation bottom [1].

Projected deformations are crucial variables for metro tunnels in operations because new construction work may affect their structural reliability and durability, as well as structural health monitoring, etc. Moreover, vertical "ovalization" (Figure 1) of a cylindrical tunnel may occur when overlying soil is removed, thus, causing unloading [2].

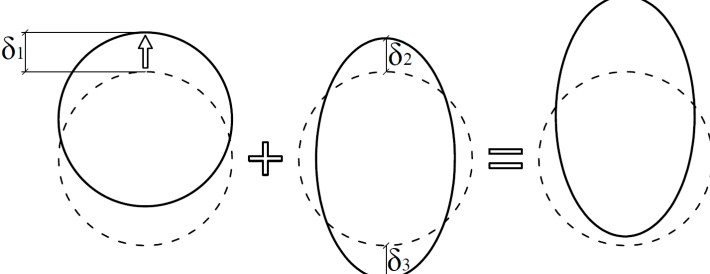

**Figure 1.** Deformations of a cylindrical tunnel due to the unloading caused by the removal of overlying soil layers.

Some studies have focused on the effect of several parameters (e.g., distance to excavation axis, excavation width, excavation length, the effect of soil improvement, etc.) on structural displacements when simulating an excavation pit adjacent to a pre-existing tunnel with a diameter of 11 m [3]. The results show that excavation work at a distance of more than 10 m from the tunnel axis has little effect on the tunnel structure, and as the length of the excavation increases, positive vertical displacements of the tunnel crown also go up. In addition, the authors of this paper mention the effect of the dissipation of excess pore pressure on the upward vertical displacement of the tunnel when the excavation work is completed, and the datum of the excavation pit bottom is reached.

Nevertheless, several deformation reduction methods can be applied during construction as tunnel protection measures. One of them improves the properties of the soil mass surrounding the tunnel by using a soil consolidation technique [3]. An increase in the thickness of the consolidated layer up to 10 m reduces vertical displacements of the tunnel to 20% [4]. Moreover, in this case, a sectional excavation method can be applied to ensure the safety of existing structures. Sectional excavation has the smallest effect on an underlying tunnel due to a reduced excavation length-to-width ratio [5]. Moreover, the arrangement of additional transverse excavation walls [6] has a positive effect; they can limit deformations caused by excavation work [5]. However, these methods boost project costs and extend the construction term.

Within the framework of such projects, a realistic geotechnical model must be developed to decide whether the above measures are necessary to ensure the safety of structures. In addition, the projected effect determines the length of the structure fragment to be monitored, the increment of the benchmarks of deformation control, and the monitoring cycle [7].

The effect of excavation work on a tunnel distributed near the excavation area has a normal distribution pattern, according to which the tunnel is almost unaffected at a distance exceeding double the depth of the excavation pit [8]. Additionally, the maximum horizontal displacement of a tunnel is insignificant compared to vertical displacement. For these reasons, it is mandatory to monitor vertical displacements within a distance of twice the tunnel length from the excavation wall. At the same time, maximum tunnel displacements are focused on the tunnel crown. Hence, the authors of this paper suggest distributing monitoring points in the upper part of the structure as well.

In the past, the authors of this paper reviewed the factors of mathematical modeling of excavation work in respect to the effect evaluation problems and considered the effect of computational assumptions on projected deformations [9]. As a follow-up study, this paper addresses the effect of the soil model, boundaries of the design scheme, and consideration of longitudinal joints in the lining (aimed at reduction in its bending stiffness) on projected additional displacements of a cylindrical tunnel below the excavation pit. The authors also performed a regression analysis of tunnel displacements depending on the following construction parameters: designed excavation pit depth, tunnel depth, and distance to the excavation pit in the plan.

One of the pressing issues in geotechnics is that geotechnical monitoring data is not used to optimize calculations due to their unavailability in the public domain, and the results of numerical modeling do not always coincide with monitoring results. This study aims to analyze the monitoring data of a metro object, obtain regression equations, and refine the parameters of mathematical modeling.

## 2. Materials and Methods

This paper focuses on the results of the automated geotechnical monitoring of a double-track metro tunnel with an outer diameter $D_{out}$ = 10.3 m, located between two designed excavation pits (Figure 2). In addition, a 24 m long tunnel is located below the bottom of an open excavation at a depth of 1.85 m to 2.4 m to the tunnel crown.

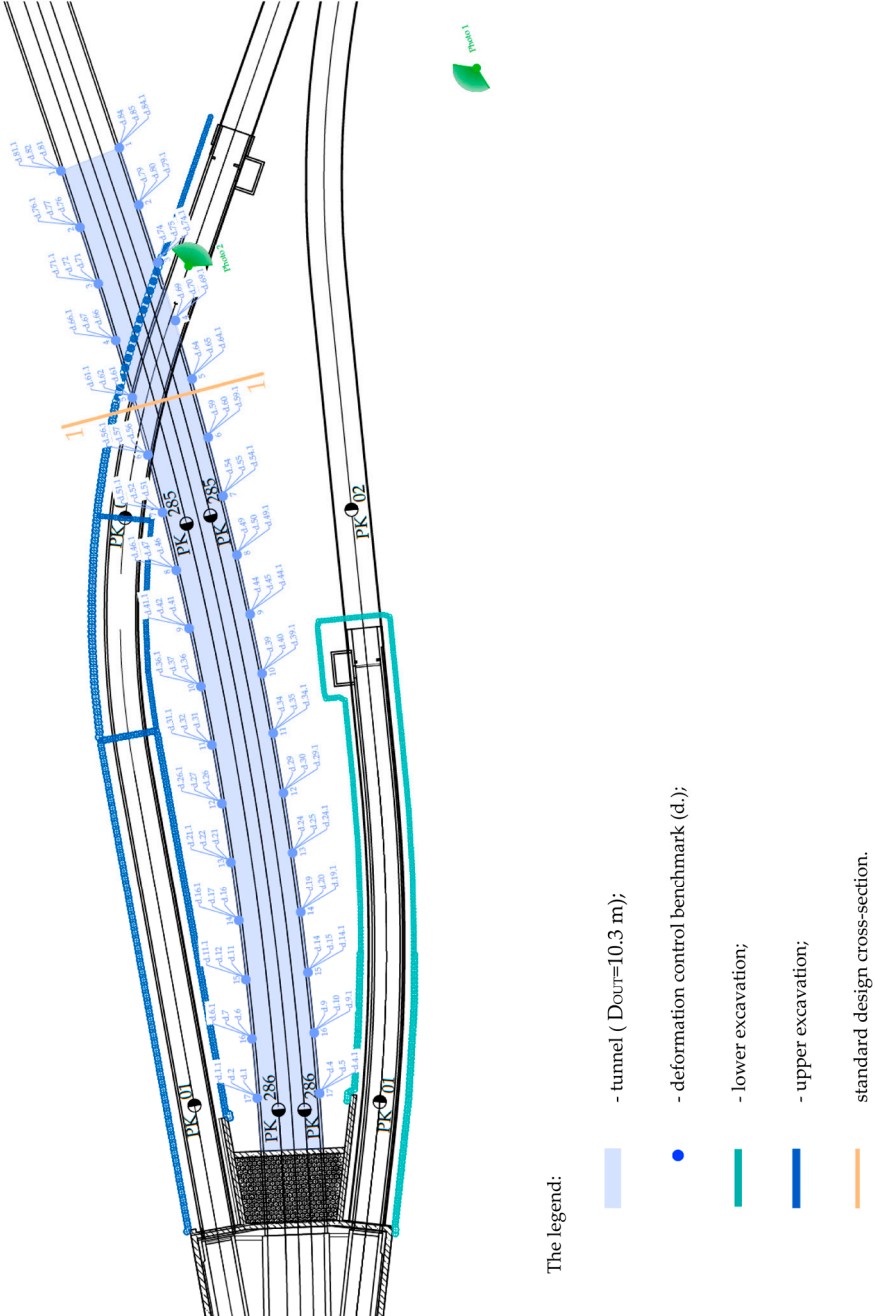

**Figure 2.** Plan with the relative location of the excavation and the tunnel with deformation control benchmarks.

This construction site was photographed while constructing an underground structure involving a reinforced concrete (RC) diagram wall and shoring (see Figures 3 and 4). The locations from which the photographs were taken are also shown in Figure 2.

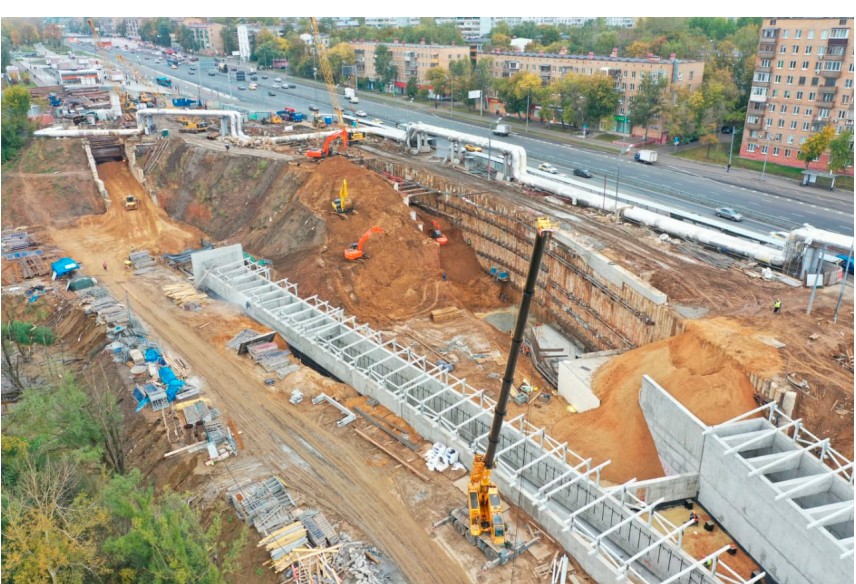

**Figure 3.** General view of the construction site (photo 1).

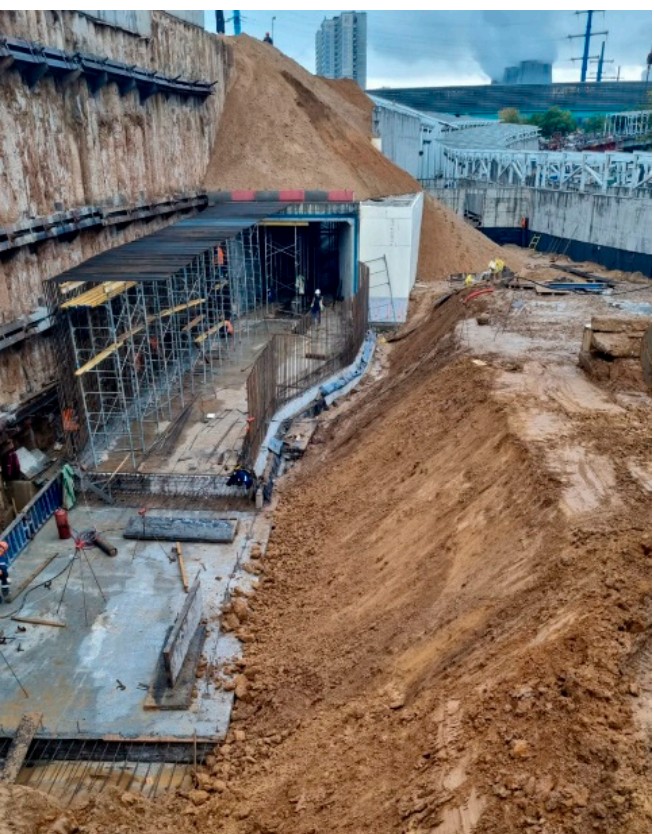

**Figure 4.** Upper excavation pit near the intersection between the designed excavation pit and the tunnel (photo 2).

Figure 5 shows the layout of geodetic signs in the cross-section of the tunnel. It is noteworthy that the absence of the deformation control benchmark in the tunnel crown is

explained by the inaccessibility of this tunnel area because of the slab for the location of the air ventilation equipment.

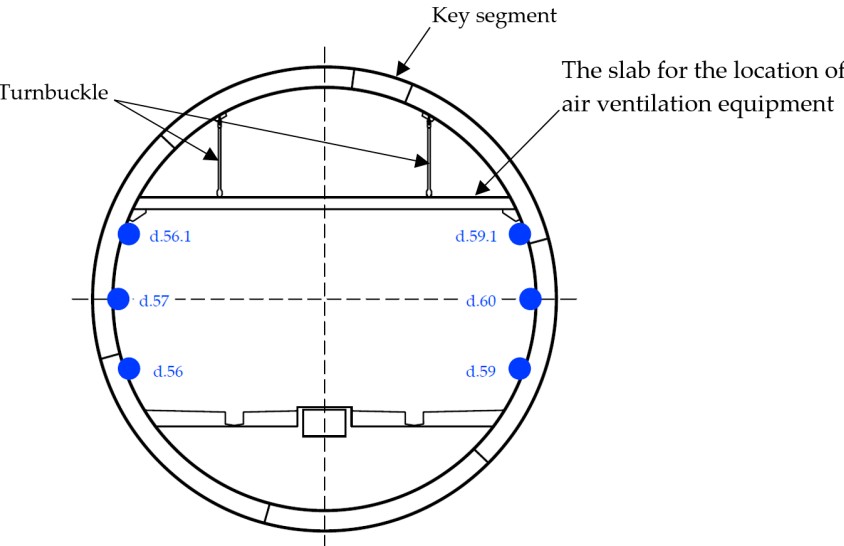

**Figure 5.** The layout of deformation control benchmarks at point 6.

The lining ring of the tunnel, affected by the new construction, consists of 7 blocks that are 0.45 m thick; one of them is a key block (Figure 5).

The considered design cross-section has fine and powdery sands followed by a semi-solid loam up to the tunnel chute.

The following soil models were used to project and compare additional and actual displacements:

- Mohr–Coulomb (MC) soil model [10];
- MC with 3 and 5-fold deformation moduli for sandy and clayey soils, respectively (3E and 5E) [11];
- Linearly elastic (LE) soil model [12];
- 3E and 5E LE models;
- Hardening Soil (HS) model [13];
- Hardening Soil Small-Strain (HSS) model [14].

The HS model, simulating nonlinear soil behaviour, and its modification HSS, are the most popular models used to solve geotechnical problems for complex construction projects. The HS model [15] is most effective for the following conditions: (1) the maximum deflection of a cantilever wall from a sheet pile is identified; (2) the settlement of facilities outside of excavation walls is identified; and (3) the maximum bending moment is determined in sandy soils. This model conveys the dependence of deformations on stresses under high loads and can take into account volumetric deformations, which are essential for hydraulic retaining structures such as rock-fill dams [16]. The findings of another study [17] demonstrate the convergence between projected results and results of the geotechnical monitoring of an embankment reinforced with mats and bamboo piles and designated for the railway infrastructure on a slope. Simulation of deep excavations in urban areas also involves HS [18] and HSS [19] models, etc.

The need to increase the deformation modulus by 3 and 5 times for sandy and clayey soil, respectively, should be explained. In Russian design practice, part of the soil mass is usually replaced under excavations to minimize soil uplift due to soil excavation in the absence of the necessary laboratory tests for HS at the preliminary evaluation stage. Thus, the findings of design schemes using the HS model simulation will be interesting for Russian designers.

Within the framework of this problem, when cross-sections are analyzed using the HSS model, stiffness parameters were taken into account for small deformations in the shear modulus ($G_0$) and the shear strain level ($\gamma_{0.7}$), according to the formulas available in the PLAXIS material modeling manual [12].

Table 1 shows the geotechnical monitoring data for vertical displacements of a tunnel during the excavation of the bottom of the upper excavation pit to the datum. During the construction period, the maximum additional displacement of the deformation control benchmark (d.56.1) was 29.4 mm.

**Table 1.** Vertical displacements of the tunnel during the excavation of the upper excavation pit: geotechnical monitoring data.

| Point № | Mark | Vertical Displacement ($S_1$), mm | Distance from the Excavation in the Plan ($r$), m | Depth of the Tunnel from the Surface to the Crown ($h$), m | Excavation Pit Depth ($H$), m |
|---|---|---|---|---|---|
| 1 | d.84.1 | −0.2 | 11.0 | 18.50 | 13.70 |
|  | d.81.1 | 0.5 | 18.8 | 11.80 | 15.60 |
| 2 | d.79.1 | 0.4 | 5.4 | 11.80 | 16.40 |
|  | d.76.1 | 0.9 | 13.3 | 2.40 | 16.40 |
| 3 | d.74.1 | 7.6 | 0.0 | 2.40 | 16.40 |
|  | d.71.1 | 4.0 | 7.5 | 2.40 | 16.30 |
| 4 | d.69.1 | 16.1 | 0.0 | 2.40 | 16.40 |
|  | d.66.1 | 12.2 | 2.0 | 2.40 | 16.60 |
| 5 | d.64.1 | 20.7 | 4.8 | 2.40 | 16.60 |
|  | d.61.1 | 22.9 | 0.0 | 2.40 | 17.10 |
| 6 | d.59.1 | 20.4 | 9.3 | 1.85 | 17.30 |
|  | d.56.1 | 29.4 | 0.7 | 1.85 | 17.60 |
| 7 | d.54.1 | 16.7 | 13.2 | 1.85 | 18.50 |
|  | d.51.1 | 24.0 | 4.1 | 1.85 | 18.50 |
| 8 | d.49.1 | 9.9 | 14.7 | 8.00 | 19.00 |
|  | d.46.1 | 15.0 | 5.2 | 8.00 | 19.00 |
| 9 | d.44.1 | 6.0 | 16.9 | 8.00 | 19.45 |
|  | d.41.1 | 10.9 | 7.2 | 8.00 | 19.45 |
| 10 | d.39.1 | 0.0 | 18.0 | 8.00 | 20.00 |
|  | d.36.1 | 4.2 | 8.3 | 8.00 | 20.00 |
| 11 | d.34.1 | −1.8 | 18.1 | 9.80 | 20.90 |
|  | d.31.1 | 2.3 | 8.6 | 9.80 | 20.90 |
| 12 | d.29.1 | −3.3 | 17.9 | 9.80 | 21.45 |
|  | d.26.1 | 0.7 | 8.3 | 9.80 | 21.45 |
| 13 | d.24.1 | −4.4 | 17.6 | 9.80 | 21.70 |
|  | d.21.1 | 0.7 | 7.9 | 9.80 | 21.70 |
| 14 | d.19.1 | −4.0 | 17.4 | 16.70 | 22.00 |
|  | d.16.1 | 0.0 | 7.6 | 16.70 | 22.00 |
| 15 | d.14.1 | −4.2 | 16.6 | 16.70 | 22.30 |
|  | d.11.1 | 0.4 | 6.9 | 16.70 | 22.30 |
| 16 | d.9.1 | −4.8 | 15.6 | 16.70 | 23.80 |
|  | d.6.1 | −0.2 | 6.0 | 16.70 | 23.80 |
| 17 | d.4.1 | −4.7 | 14.6 | 16.70 | 24.25 |
|  | d.1.1 | −1.1 | 5.0 | 16.70 | 24.25 |

In this paper, the effect of excavating a shallow pit is not taken into account, since the excavation of one fragment of a pit had almost no effect on displacements, while the second fragment of the lower excavation pit is relatively shallow, and it had no effect on additional deformations.

Table 2 shows horizontal displacements ($S_2$). Maximum horizontal displacements of the tunnel are registered both in the horizontal diameter of the tunnel, or the spring line, and just above it, under the air slab (marks having numbers that end in ".1"). For this reason, the most significant displacement at each point is included in the table.

**Table 2.** Horizontal displacements of the tunnel during the excavation of the upper excavation pit: geotechnical monitoring data.

| Point № | Mark | Horizontal Displacement ($S_2$), mm | Distance from the Top ($r$), m | Depth of the Tunnel from the Surface to the Crown ($h$), m | Depth of the Upper Excavation Pit ($H$), m | Distance from the Bottom of the Excavation Pit to the Bottom of the Tunnel ($d$), m |
|---|---|---|---|---|---|---|
| 1 | d.84.1 | 1.3 | 11.0 | 18.50 | 13.70 | 10.00 |
| | d.81.1 | 1.1 | 18.8 | 11.80 | 15.60 | 10.00 |
| 2 | d.79.1 | 2.2 | 5.4 | 11.80 | 16.40 | 10.00 |
| | d.76.1 | 0.9 | 13.3 | 2.40 | 16.40 | 10.00 |
| 3 | d.74.1 | 3.4 | 0.0 | 2.40 | 16.40 | 10.20 |
| | d.71.1 | 1.0 | 7.5 | 2.40 | 16.30 | 10.20 |
| 4 | d.70 | −5.1 | 0.0 | 2.40 | 16.40 | 11.80 |
| | d.66.1 | 3.5 | 2.0 | 2.40 | 16.60 | 11.80 |
| 5 | d.64.1 | −9.8 | 4.8 | 2.40 | 16.60 | 13.80 |
| | d.62 | 2.4 | 0.0 | 2.40 | 17.10 | 13.80 |
| 6 | d.59.1 | −11.4 | 9.3 | 1.85 | 17.30 | 13.00 |
| | d.56.1 | 2.6 | 0.7 | 1.85 | 17.60 | 13.00 |
| 7 | d.54.1 | −8.5 | 13.2 | 1.85 | 18.50 | 9.30 |
| | d.51.1 | −7.0 | 4.1 | 1.85 | 18.50 | 9.30 |
| 8 | d.49.1 | −5.0 | 14.7 | 8.00 | 19.00 | 9.50 |
| | d.46.1 | −9.8 | 5.2 | 8.00 | 19.00 | 9.50 |
| 9 | d.44.1 | −2.1 | 16.9 | 8.00 | 19.45 | 9.30 |
| | d.41.1 | −10.8 | 7.2 | 8.00 | 19.45 | 9.30 |
| 10 | d.39.1 | 5.3 | 18.0 | 8.00 | 20.00 | 9.30 |
| | d.37 | −6.4 | 8.3 | 8.00 | 20.00 | 9.30 |
| 11 | d.34.1 | 11.4 | 18.1 | 9.80 | 20.90 | 6.50 |
| | d.32 | −3.1 | 8.6 | 9.80 | 20.90 | 6.50 |
| 12 | d.29.1 | 13.3 | 17.9 | 9.80 | 21.45 | 6.00 |
| | d.27 | 3.8 | 8.3 | 9.80 | 21.45 | 6.00 |
| 13 | d.24.1 | 13.1 | 17.6 | 9.80 | 21.70 | 5.50 |
| | d.22 | −5.8 | 7.9 | 9.80 | 21.70 | 5.50 |
| 14 | d.19.1 | 12.3 | 17.4 | 16.70 | 22.00 | 5.00 |
| | d.16.1 | 4.4 | 7.6 | 16.70 | 22.00 | 5.00 |
| 15 | d.14.1 | 9.9 | 16.6 | 16.70 | 22.30 | 4.70 |
| | d.12 | −4.8 | 6.9 | 16.70 | 22.30 | 4.70 |
| 16 | d.9.1 | 5.6 | 15.6 | 16.70 | 23.80 | 3.20 |
| | d.6.1 | −8.9 | 6.0 | 16.70 | 23.80 | 3.20 |
| 17 | d.05 | 2.6 | 14.6 | 16.70 | 24.25 | 2.68 |
| | d.02 | −10.8 | 5.0 | 16.70 | 24.25 | 2.68 |

Multiple regression models [20] were made using the data, provided in Tables 1 and 2. Graphs were also made to demonstrate the interaction between vertical and horizontal tunnel displacements with other parameters (Figures 6 and 7). The coefficients of linear regression were obtained using an Excel add-in called Data Analysis. Simulation results are presented in the next section.

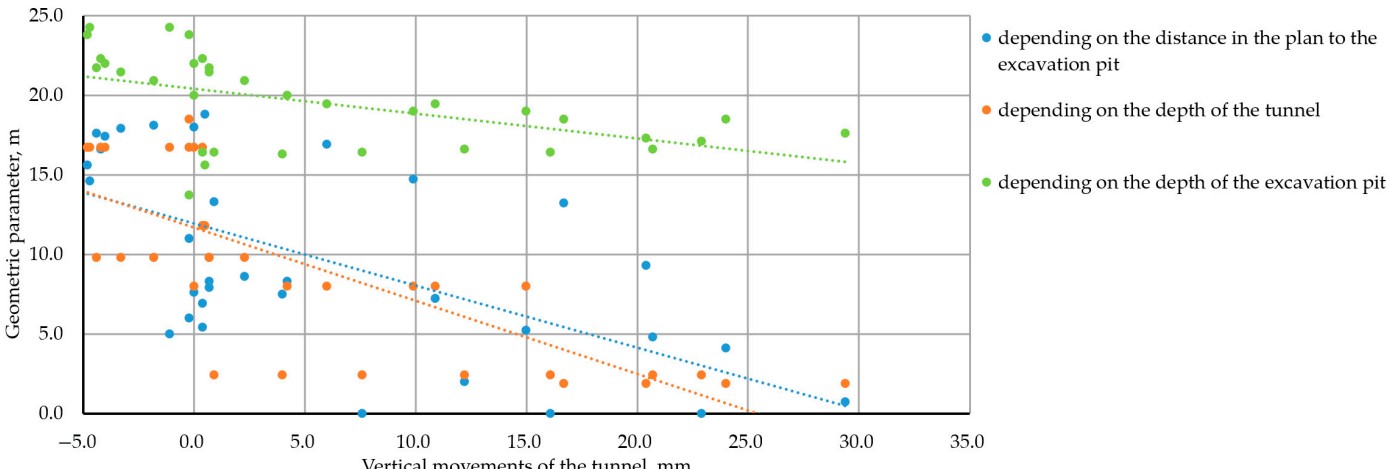

**Figure 6.** Vertical displacements of the tunnel depending on the parameters.

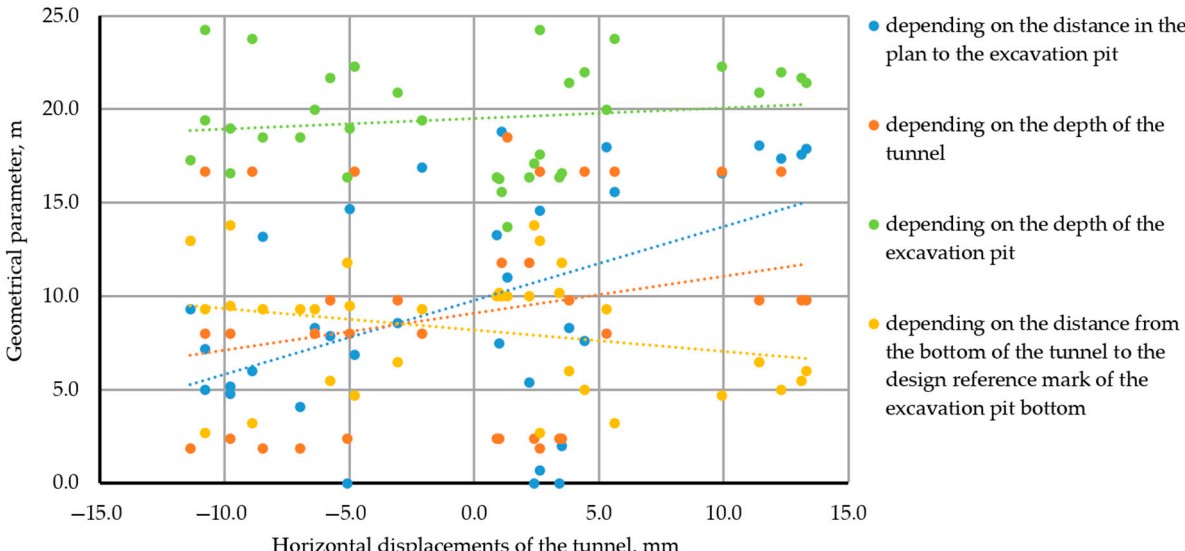

**Figure 7.** Horizontal displacements of the tunnel depending on the parameters.

Previously, multiple regression equations were made by applying the finite element method (FEM). These equations were made for maximum and horizontal displacements of soil massifs outside the excavation pit, and they involved the use of 144 design schemes [21]. The equations consider the width and depth of the excavation pit, the shear strength of the soil, and the stiffness of the excavation wall and the stull-set system. According to this study's results, values of the coefficient of determination $R^2$ [22] were 0.95 for each of the equations, which means that this equation can predict the deformation splendidly.

In addition, the design scheme, describing the tunnel located under the excavation pit, was subjected to backward analysis to clarify the computational assumptions of mathematical modeling disregarding the groundwater lowering due to its absence on the construction site. However, this needs to be taken into account during modeling, as it affects the increase in effective vertical stress in the soil, and as a result, increases the soil displacement [23].

Computational models had different distances from the tunnel bottom to the bottom of the design scheme: $0.5D_{out}$ and $0.25D_{out}$. The Muir–Wood (M-W) coefficient [24] was introduced in some schemes to consider the joints between lining blocks made to reduce the bending stiffness of the concrete lining. According to this methodology, the lining is specified as continuous and has no joints, and it applies to this number of blocks [25].

The results of the schemes elaboration are also presented in the next section. A standard design section is shown in Figure 8 at the stage of excavation to -> the datum where the excavation bottom is located.

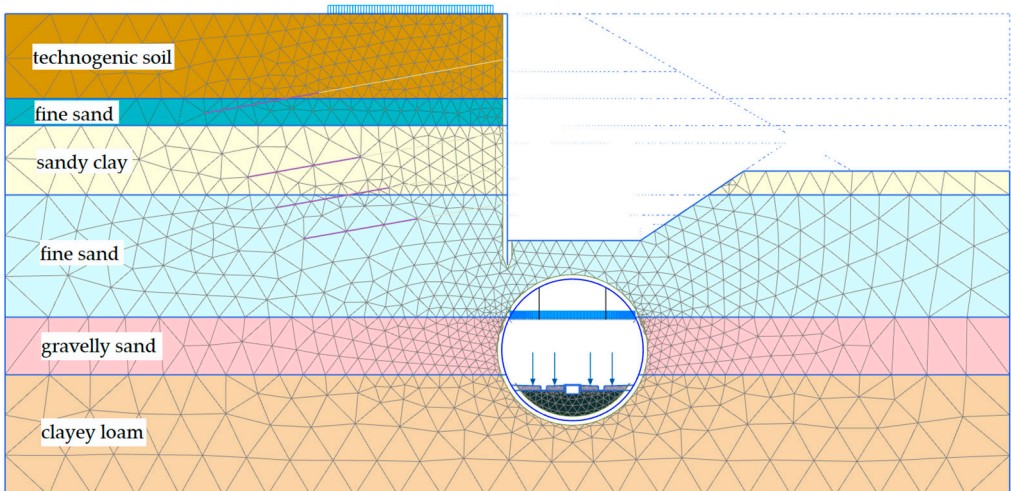

**Figure 8.** A standard 1-1 section at the time of excavation to the datum of the bottom of the designed excavation pit at the point of intersection between the designed and the existing structure.

## 3. Results

Vertical displacement ($S_1$) is a dependent variable depending on the following:

— distance from the excavation pit in the plan ($r$);
— depth of the tunnel from the surface to the crown ($h$);
— depth of the excavation pit ($H$).

If the data obtained by the authors are taken into account, the multiple regression equation, describing vertical displacements of a tunnel, is as follows:

$$S_1 = 23.48 - 0.58{\cdot}r - 0.97{\cdot}h - 0.17{\cdot}H \tag{1}$$

Regression coefficients of variables $r$ and $h$ are significant, as evidenced by the obtained *p*-values, which are below the level of significance $\alpha = 0.05$ [26], or $P_r = 0.0024 < 0.05$, and $P_h = 0.0001 < 0.05$. At the same time, the *p*-value of the excavation pit depth parameter ($H$) is $P_H = 0.701 \gg 0.05$. However, we cannot eliminate the summand with this parameter from the equation, because the correlation analysis [27], whose results are presented in Table 3, demonstrates the interrelationship with parameters $S_1$ and $h$.

**Table 3.** Matrix of pairwise correlation coefficients of parameters used in the multiple regression model.

| Parameter | $S_1$ | $r$ | $h$ | $H$ |
|-----------|-------|------|------|-----|
| $S_1$ | 1 | - | - | - |
| $r$ | −0.62 | 1 | - | - |
| $h$ | −0.77 | 0.41 | 1 | - |
| $H$ | −0.53 | 0.32 | 0.62 | 1 |

The significance of *F*, used to check the joint significance of all criteria of the equation [28], is far below the significance level $\alpha = 0.05$, i.e., $F_{S1} = 0.00000008 << 0.05$. This test result indicates the high reliability of the solution.

Moreover, for this equation, the multiple correlation coefficient R is equal to $R_{S1} = 0.84$ [27], and the coefficient of determination $R^2$, which is equal to $R^2_{S1} = 0.70$, measures how well a statistical model predicts the outcome [22].

The equation of horizontal displacements ($S_2$) has an additional parameter (*d*), and it consists of the values of points 17 to 7, inclusively (see Table 2 and Figure 2). In this range, this equation is characterized by significance $F_{S2} = 0.00011444 << 0.05$:

$$S_2 = 99.72 + 1.28 \cdot r - 0.30 \cdot h - 4.44 \cdot H - 3.69 \cdot d \tag{2}$$

In this equation, parameter r is the most significant ($P_r = 0.004849198 < 0.05$). In the regression equation describing horizontal displacements, multiple correlation coefficient R equals $R_{S_2} = 0.85$ [29], while the coefficient of determination equals $R^2_{S_2} = 0.73$ [22].

It is found that the accuracy of the regression equation for horizontal displacements decreases as the points from 6 to 1 are added stepwise. For example, point 6, if added, causes a 100-fold reduction in accuracy, and coefficients $R_{S_2}$ and $R^2_{S_2}$ cause a decrease of about 21 and 50%, respectively. Hence, the authors apply the multiple regression equation only to that part of the tunnel conventionally parallel to the excavation pit before the intersection between the existing and designed structures.

Table 4 shows the results of the simulation of design schemes using different soil models, the bending stiffness of the lining, and the distance from the tunnel to the bottom of the design scheme. Figure 9 illustrates simulation results in the order of decreasing prognosticated values.

**Table 4.** Consolidated simulation results.

| Scheme № | Soil Model | Distance from the Bottom of the Tunnel to the Bottom of the Design Scheme (*l*) *, m | M-W Coefficient Taken (+)/Not Taken (−) into Account | Tunnel Displacement at the Stage of Excavation to the Design Reference Mark of the Excavation Pit Bottom, mm | | |
|---|---|---|---|---|---|---|
| | | | | Model 1 (Point 5, d.61.1) | Model 2 (Point 6, d.56.1) | Model 3 (Point 7, d.51.1) |
| 0 ** | - | - | − | 22.9 | 29.4 | 24.0 |
| 1 | LE | 5.2 | + | 150.7 | 146.8 | 133.3 |
| 2 | LE with 3E and 5E | 5.2 | − | 72.0 | 69.2 | 56.4 |
| 3 | MC | 5.2 | + | 184.7 | 182.3 | 163.8 |
| 4 | MC with 3E and 5E | 5.2 | + | 79.9 | 71.6 | 58.6 |
| 5 | MC | 2.6 | + | 124.6 | 120.3 | 102.1 |
| 6 | MC with 3E and 5E | 2.6 | + | 55.8 | 48.1 | 37.2 |
| 7 | MC | 5.2 | − | 180.0 | 176.7 | 152.5 |
| 8 | MC with 3E and 5E | 5.2 | − | 78.3 | 68.9 | 57.1 |
| 9 | MC | 2.6 | − | 121.9 | 117.3 | 98.2 |
| 10 | MC with 3E and 5E | 2.6 | − | 49.8 | 43.2 | 34.6 |
| 11 | HS | 5.2 | + | 28.0 | **27.4** | **22.1** |
| 12 | HS | 5.2 | − | 26.6 | 26.8 | 21.1 |
| 13 | HSS | 5.2 | + | 23.8 | 21.6 | 17.6 |
| 14 | HSS | 5.2 | − | **22.8** | 20.8 | 16.8 |

* Distance from the bottom of the excavation pit to the bottom of the design scheme *L* = *l* + 12.5 m; ** Results of field observations are provided for clarity purposes; The displacements closest to the field observations are highlighted in bold.

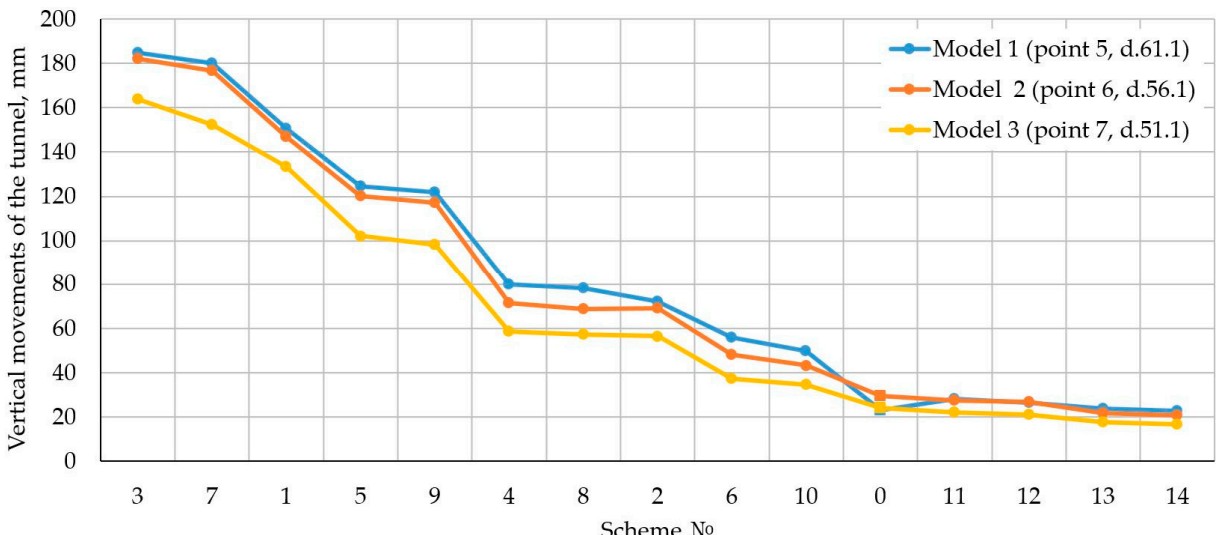

**Figure 9.** Displacements of the tunnel during excavation down to the datum of the excavation bottom, depending on the soil model, the bending stiffness of the lining, and the boundary conditions.

Analysis of projectable results and automated geotechnical monitoring data allows for the conclusion that a combination of (1) the HS soil model, (2) the M–W coefficient, and (3) the $0.5D_{out}$ distance from the bottom of the tunnel to the bottom of the design scheme in models 2 and 3 demonstrate the best convergence with the field observations, with an error below 9%. Hence, scheme 11 appears to be closest to the findings of automated tunnel monitoring performed during construction. Meanwhile, for model 1, scheme 14 with the HSS soil model simulates the structural behaviour closest to the actual behaviour of constructions with the distance of $0.5D_{out}$ from the tunnel bottom to the bottom of the design scheme, if the M–W coefficient is ignored.

## 4. Discussion

This paper considers the effect of the soil model, the joints between lining blocks using the Muir–Wood coefficient, and the lower boundary of the design scheme on the resulting displacement of the tunnel located below the designed excavation and the convergence with the results of geotechnical monitoring obtained during construction.

The authors agree with other researchers [15–19] that the most accurate prediction, which is close to actual observations, is found in the schemes that use the HS model and its HSS modification. The authors found that the Muir–Wood coefficient affects an increase in vertical displacements of the tunnel towards the excavation work. The increase ranges from 0.6 mm to 11.3 mm (on average by 4.8%), as the lining becomes more susceptible to deformations. The results of calculations with and without this coefficient are presented in Table 4.

In addition, a multiple regression equation was proposed for vertical displacements depending on the tunnel location relative to the excavation, including the tunnel location below the designed structure. A multiple regression equation for horizontal displacements of the tunnel part parallel to the open excavation was also proposed.

The determination coefficients of the compiled equations, $R^2$, are slightly smaller than the accuracy of regression equations in the study mentioned above [21] (by about 26%). The obtained result indicates that not all factors affecting the deformations were taken into account. However, this work is based on the actual data of the geotechnical monitoring of a circular tunnel during the construction of new metro facilities, rather than modeling results, which is an undoubted advantage. Nevertheless, these equations can be used to make a preliminary evaluation of the deep underground structure's displacement caused by the upper pit excavation, since these equations contain easily obtained geometric parameters (distance in plan, depth, etc.), which are available at the initial design stage.

The authors plan to continue the study to determine more factors affecting tunnel deformations to increase the significance of regression equations.

It should also be noted that the air slab makes it challenging to access the tunnel crown to perform geotechnical monitoring. However, due to the vertical "ovalization" of the structure, the serviceability of the structure and control of the maximum deflection of its span must be prioritized among all other tasks for in situ tunneling surveyors. Hence, it is recommended to place a deformation control mark in the middle of the slab span for tunnels with an outer diameter of 10.3 m made using a similar design solution.

**Author Contributions:** Conceptualization, methodology, V.P.K.; software, validation, I.O.I. and V.V.R.; formal analysis, writing—review and editing, all authors; investigation, A.Z.T.-M., I.O.I. and V.V.R.; supervision, project administration, A.Z.T.-M.; resources, V.P.K. and I.O.I.; visualization, V.V.R. All authors have read and agreed to the published version of the manuscript.

**Funding:** This work was financially supported by the Ministry of Science and Higher Education of Russian Federation (grant # 075-15-2021-686). Tests were carried out using research equipment of The Head Regional Shared Research Facilities of the Moscow State University of Civil Engineering.

**Data Availability Statement:** Not applicable.

**Conflicts of Interest:** The authors declare no conflict of interest.

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
