# Peer review of "Projected Effects of a Deep Excavation Pit on the Existing Metro Tunnel and Findings of Geotechnical Monitoring: A Comparative Analysis"

_buildings, doi:10.3390/buildings13051320_

Round 1

Reviewer 1 Report

Review of the article buildings-2379530
Projected effects of a deep excavation pit on the existing metro
tunnel and findings of geotechnical monitoring: comparative
analysis

In the paper, the authors presented a retrospective analysis of the design scheme developed for the tunnel construction site over the existing tunnel with a diameter of 10.3 m based on the results of geotechnical monitoring performed in PLAXIS 2D. The optimal combination of the distance from the bottom of the tunnel to the lower boundary of the design model, the soil model and the rigidity of the tunnel lining was revealed. The authors have given regression equations describing the vertical and horizontal displacements of the tunnel at the stage of excavation to the design control level of the bottom of the excavation. The studies are of practical importance in the construction of the relevant facility, but for publication it is necessary to clarify the following issues:

1. The paper presents multiple regression equations, but it is not indicated how they were obtained: 1) using the author's calculations; 2) using software packages, which ones?

2. The term forecasting is often found in the title of the article and in the text, but in fact there is no commentary on how it is obtained, what formulas are used to calculate.

3. Section 4 provides statistics in % and links to tables where percentages are missing.

4. According to the text of the article, the significance of the regression equation is identified with the accuracy of the regression equation itself. These are completely different concepts in fact.

5. Section 4, segment 4. The coefficient of determination of the multiple regression equation and the significance of the multiple regression equation are different concepts. They cannot be compared.
Conclusion. For publication, revision of the text of the article is required. Reviewer.

Reviewer 2 Report

1. Two reviewing reports are attached. One is a grammar report showing 90 out of 100 for English presentation. Therefore, minor editing of English may be required with the assistance of these the grammar report attached. Grammarly is recommended to assist with your English proofing at the initial stage.

2. The other is a similarity report showing acceptable similarity to previously published work. However, there is a 4% similarity to an MDPI publication, as noted in the similarity report. Please consider reducing the similarity regarding this 4%.

3. Please edit your mathematical equations using Word equation editor or Mythtype, or any other mathematical equations editing software. Also, all mathematical symbols should be in italics. 

4.  Please leave a space between any math symbol and values as well as equations (x = a + b) and inequation (x < 1; X > 3). There should be no space for a less or large than a value (standard deviation = ±2; <1; >2).

5. Please scrutinize your main text using Grammarly or Writefull until there are no more typos and misspellings.

6. Please improve your data charts in Excel. For instance, black colour for x- and y-labels and axis, larger font size for labels and axis values, dark colour for x and y axes, and more remarkable colour for data points and fitting lines. 

7. Table 3 has all math symbols in italics but is not done for any other tables. Please keep the consistency of all math symbols in italics.

8. Despite word connection, please replace all hyphens with a minus (e.g., "stress-induced" using a hyphen is correct, 3 – 1  = 2 using a minus is correct).

9. The introduction could be written in a funnel-shaped logic from a broader engineering background, narrowing down to a specific research gap and objective.

10. The conclusion could be summarized in a few points rather than lumped into several paragraphs, which reduces the reliability and clarity for readers.

11. Please return this manuscript back with line numbers. Otherwise, any specific comment cannot be provided for each line regarding writing, technical and academic quality.

Please return this manuscript back with line numbers. Otherwise, any specific comment cannot be provided for each line. 

Round 2

Reviewer 1 Report

Повторное рецензирование статьи buildings-2379530

Проектируемый Влияние глубокого котлована на существующее метро

туннель и Результаты геотехнического мониторинга: сравнительный анализ

Ответ авторов был следующим: проанализирован и рассмотрен доработанный вариант статьи. Я согласен с Комментарии авторов. Доработанный вариант рекомендуется к публикации.

          Рецензент.

Author Response

Thank you a lot for your review of our article.

Reviewer 2 Report

See the file attached.

It is amendable by an English-proofing editor once accepted.

Round 3

Reviewer 2 Report

Consent to publish as it is.

96 out of 100 for Grammarly report check. Well done. 

The English proofing editor will help them publish this work after English proofreading.